# RNase III, Ribosome Biogenesis and Beyond

**DOI:** 10.3390/microorganisms9122608

**Published:** 2021-12-17

**Authors:** Maxence Lejars, Asaki Kobayashi, Eliane Hajnsdorf

**Affiliations:** 1Division of Biomedical Science, Faculty of Medicine, University of Tsukuba, Tsukuba 305-8575, Japan; asaki.lejars@md.tsukuba.ac.jp; 2Institut de Biologie Physico-Chimique, UMR8261, CNRS, Université de Paris, 75005 Paris, France

**Keywords:** RNase III, ribosome biogenesis, bacteria, eukaryotes

## Abstract

The ribosome is the universal catalyst for protein synthesis. Despite extensive studies, the diversity of structures and functions of this ribonucleoprotein is yet to be fully understood. Deciphering the biogenesis of the ribosome in a step-by-step manner revealed that this complexity is achieved through a plethora of effectors involved in the maturation and assembly of ribosomal RNAs and proteins. Conserved from bacteria to eukaryotes, double-stranded specific RNase III enzymes play a large role in the regulation of gene expression and the processing of ribosomal RNAs. In this review, we describe the canonical role of RNase III in the biogenesis of the ribosome comparing conserved and unique features from bacteria to eukaryotes. Furthermore, we report additional roles in ribosome biogenesis re-enforcing the importance of RNase III.

## 1. Introduction

Universally conserved, the ribosome is a complex ribonucleoparticle that acts as the catalyst for protein synthesis. Ribosomal RNAs (rRNAs) and ribosomal proteins are assembled in a stepwise manner involving a plethora of protein and RNA effectors. As one of the most elaborate biological machines, the ribosome remains a major object of study in regard to its complex nature and its heterogeneity in cells [1]. In prokaryotes and eukaryotes, ribosomal biogenesis relies on the transcription of precursors rRNAs, which are processed into mature rRNAs [2,3]. The complexity of the step-by-step process of maturation reveals a need for precision in the timing as well as robustness in the synthesis of rRNAs. In particular, rRNAs adopt complex structures, which are achieved through double-stranded RNA (dsRNA) motifs [4] and which are essential for both catalytic and structural integrity of the mature ribosome [5].

In cells, RNAs are protected by RNA-binding proteins (RBP) and RNA chaperones and are processed by ribonucleases (RNases). Although naked single-stranded RNAs can be cleaved by various RNases, the processing of stable dsRNA structures requires specialized RNase III domain (RIIID)-containing enzymes. RNase IIIs are endoribonucleases cleaving dsRNA conserved from bacteria (e.g., RNase III) to eukaryotes (e.g., Rnt1, Drosha and Dicer) both in terms of structure and function [6,7,8,9,10,11]. RNase III was first identified for its role in the initial step of rRNA maturation in the model organism *Escherichia coli* [12]. Subsequent studies demonstrated that RNase III is also involved in the regulation of gene expression in bacteria and the maturation of non-coding RNAs in eukaryotes.

In this review, we focus on the diversity of RNase III enzymes from bacteria to eukaryotes and their roles in ribosome assembly. First, we illustrate the diversity in the RNase III family from *E. coli* RNase III to the more specialized Drosha and Dicer enzymes. We then recapitulate how RNase III enzymes are directly involved in the maturation of rRNAs. Finally, we report other indirect roles of RNase III in ribosome biogenesis, highlighting its importance in ribosome homeostasis.

## 2. The Founding Member

### 2.1. The rnc Gene

RNase III was discovered in *E. coli* (referred hereafter as Ec-RNase III) in 1967 [13] and shown to cleave natural and synthetic dsRNAs in vitro in 1968 [14]. Isolation of an RNase III-deficient derivative of *E. coli* in 1973 [15] revealed its importance in the initial steps of the maturation of rRNA precursors [12]. Assuming that Ec-RNase III could have an important role in cell physiology, Kindler et al. isolated thermosensitive mutants from an *E. coli* strain inactivated for RNase I [16]. The clone AB301-105 carrying the *rnc105* mutation (Table 1) demonstrated a strongly reduced ability to cleave dsRNA substrates in vitro [15]. The *rnc105* mutation was then mapped around 49 min on the chromosome of *E. coli*, identifying the *rnc* gene, which was subsequently confirmed to encode Ec-RNase III [17]. Although Ec-RNase III is not essential for growth [18], its inactivation was found to provoke slow growth and sensitivity to heat [19] and osmotic shock [20].

Ec-RNase III is co-expressed with the essential GTPase Era within the *rnc*-*era-recO-pdxJ-acpS* operon. The *recO* gene encodes a DNA repair protein while the *pdxJ* and *acpS* genes, also transcribed from their own promoters, encode the pyridoxine synthase PdxJ, essential for growth in the absence of pyridoxine and the essential holo-acyl-carrier-protein synthase AcpS, respectively. Hence, as demonstrated by the absence of an *rnc* deletion mutant in the Keio collection [34], inactivating Ec-RNase III likely lead to polar effects on the expression of the gene *era*, which was shown to be essential but is also toxic when overexpressed [35]. Interestingly, insertion mutants of RNase III (*rnc14* and *rnc38,* carrying large inserts containing transposons and fragments of plasmid) seem to provide the correct compensation of *era* expression. In addition, RNase III cleaves its own mRNA in its 5′ untranslated region (UTR), leading to the destabilization of both *rnc* and *era* mRNAs (see Section 2.2.3) [36,37].

Bacterial RNase III carry two distinct domains: an N-terminal region containing the characteristic catalytic core of RNase III (RIIID) (amino acids 6–128) and the double-stranded RNA (dsRNA)-binding domain (dsRBD) in the C-terminal region (amino acids 155–225) (Figure 1). Various mutations were characterized, which disrupt the catalytic activity of Ec-RNase III (with or without affecting its binding affinity) or abolish its expression (Table 1). The mutations which affect catalytic ability are all located in the RIIID with the exception of *rnc7* (D155E) and Q153 located in the dsRBD and the linker, respectively [31]. Importantly, as shown for a few mutants (e.g., *rnc70*), the catalytic activity of RNase III can be lost without affecting its binding to dsRNA. Moreover, the *rnc70* mutant was shown to be dominant over the wt allele in the regulation of the *N* mRNA from the lambda phage, suggesting that dsRNA binding and cleavage are independent functions of Ec-RNase III [30].

### 2.2. How Does RNase III Work?

#### 2.2.1. Structure

The dimerization of the active Ec-RNase III complex relies on the interaction of two RIIIDs to form a single processing center accommodating two divalent metallic ions, with a preference for magnesium [29]. The interaction of the two RIIIDs, achieved through a ball and socket junction, is essential for the correct positioning of the cleavage sites [23]. In the UniProt database (https://www.uniprot.org [38], accessed on the 11 October 2021), RNase III structures from various bacterial species (i.e., *Aquifex aeolicus* O67082, *Campylobacter jejuni* Q9PM40, *Thermotoga maritima* Q9X0I6 and *Mycobacterium tuberculosis* P9WH03) are available but not from *E. coli*. These structures revealed that the RIIID is composed of seven α-helices and adopts a unique fold while the dsRBD adopts an α-β-β-β-α topology, as found in other dsRBD containing enzymes [39]. While no complete structure of Ec-RNase III is currently available, Ec-RNase III dsRBD was solved by NMR spectroscopy showing a typical α-β-β-β-α topology [40]. Furthermore, structure prediction performed recently through the AlphaFold program (https://alphafold.ebi.ac.uk, accessed on the 11 October 2021) [41,42] supports a common structure between Ec-RNase III (P0A7Y0) and previously obtained RNase III structures in bacteria. In addition, important amino acid residues have been identified from a wide range of Ec-RNase III mutants (Table 1) and include residues E38, E41, F42, G44, D45, E65, G97, E100, D114, E117 in the RIIID, Q153 in the linker and D155 and A211 in the dsRBD. In particular, negatively charged residues E38, E41, D45, E65, E100, D114 and E117 are highly conserved among bacterial RNase III enzymes [21] (Figure 1).

#### 2.2.2. Mechanism

Recruitment of an RNase III dimer is initiated by the interaction of a single dsRBD with a target dsRNA. The binding induces a conformational change of the dsRBD, bringing it closer to the dsRNA which facilitates the symmetric recruitment of the second dsRBD [43]. Upon binding, the second dsRBD is reoriented which renders the catalytic core functional. Of note, if the dsRNA structure does not correctly accommodate the dimer, a non-catalytic RNase III complex can form and the regulatory outcome of this interaction remains unclear. At the molecular level, RNA cleavage relies on a nucleophilic attack, allowing the hydrolysis (supposedly simultaneous) of the phosphodiester bond generating 3′-hydroxyl and 5′-phosphate ends on both strands but which are staggered by two bases on one strand of the dsRNA compared to the other (Figure 2). The release of the generated cleavage products remains poorly characterized but was shown to be the limiting step of the reaction likely due to the requirement for a second change in the conformation of the dimer to release the processed RNA fragments [44].

If the early studies of Ec-RNase III suggested that binding and (single or double-stranded) cleavage was not random, criteria for RNase III target selection remain poorly understood. In the pre-mRNA of the T7 phage, multiple single-stranded cleavages sites were identified, of which the “R1.1” cleavage, located at the 5887th nucleotide (nt) of the T7 phage pre-mRNA, drives the maturation and protection of T7 mRNAs (Figure 2) [45,46]. R1.1 is formed by a 49 nts long stem carrying symmetrical sets of proximal (relative to the cleavage site) boxes (pb, 4 nts), middle boxes (mb, 2 nts) and distal boxes (db, 2 nts) for each of the monomers to bind the dsRNA [11,47]. R1.1 became a model to understand the properties of RNase III binding and cleavage and a mutant named R1.1 WC-L, in which full complementarity is forced in the stem enabled the double-stranded cleavage of the RNA [29]. Shortening of the R1.1 stem revealed that a single set of boxes in a 12 base pair (bp) stem with a tetraloop is sufficient for single-strand cleavage by Ec-RNase III (µR1.1 Δ1) while the cleavage is lost upon removal of the last bp of the distal box (µR1.1 Δ2) [48]. It is noteworthy that the recent identification of thousands of staggered double-stranded Ec-RNase III cleavage sites in *E. coli* suggests that RNase III binding sites do not share a strong consensus sequence but likely depend on the structure of the targeted stem loop [49].

#### 2.2.3. Physiological Roles of RNase III

In addition to its role in rRNA maturation, the importance of bacterial RNase III was highlighted through transcriptomic studies performed in diverse organisms, including but not limited to *E. coli* [49,50,51,52], *Streptomyces coelicolor* [53], *Staphylococcus aureus* [54], *Synechococcus* sp. PCC7002 [55] or *Rhodobacter sphaeroides* [56]. These studies demonstrate the pleiotropic role of RNase III in the control of gene expression and a comparison of the genes affected by RNase III inactivation in these organisms would be informative about the distribution and conservation (or not) of targets. In *E. coli*, RNase III is involved in the destabilization of numerous RNAs. For example, Ec-RNase III cleaves its own mRNA in the 5′UTR of the *rnc*-*era* mRNA, which destabilizes the whole transcript (Figure 3) [37]. Ec-RNase III can cleave in between coding sequences within polycistronic mRNAs such as *rpsO*-*pnp*, encoding the ribosomal protein S15 and the exoribonuclease PNPase. This cleavage leads to the destabilization of *pnp* mRNA without affecting the expression of *rpsO* (Figure 3). Cleavages within coding sequences were also found as in the *arfA* mRNA (Figure 3), encoding the alternative ribosome rescue factor ArfA, thus revealing a positive role of Ec-RNase III in an alternative pathway to rescue stalled ribosomes upon mRNA truncation [57]. While targets of RNase III in bacteria are usually expected to be negatively regulated, maturation can also lead to positive regulation as in the case of the pre-rRNA (see Section 4.1). RNase III is also involved in intermolecular dsRNA cleavages (i.e., where the dsRNA is composed of two distinct molecules) as in the case of regulatory RNAs bound to their targets. For the small RNA RhyB, binding to the *sodB* mRNA, encoding the superoxide dismutase FeSOD, RNase III cleavage leads to the degradation of both RNAs [58] while, on the other hand, the cleavage of the antisense RNA (asRNA) ArrS bound to the *gadE* mRNA leads to increased translation of GadE, an acid response transcriptional factor [59]. These and other examples demonstrate the pleiotropic functions of RNase III in bacterial physiology. For example, in the adaptation phase following an osmotic shock RNase III activity is repressed, which allows stabilization of *proP* (Figure 3), *proU* and *betT* mRNAs encoding proteins involved in the import of osmoprotectants [60,61,62]. Furthermore, RNase III was shown to be involved in thermotolerance [19], motility [63] and aminoglycoside resistance [64]. In other bacteria, RNase III was shown to be important for a whole range of functions, including but not limited to methionine biosynthesis in *S. aureus* [65] and cell wall homeostasis in *Pseudomonas putida* [66] and to be involved in virulence in *Enterococcus faecalis* [67], *Listeria monocytogenes* [68], *S. aureus* [54] or *Campylobacter jejuni* [69].

Although the majority of characterized RNase III target sites in *E. coli* likely result from intramolecular dsRNA, it was recently shown that among the thousands of in vivo Ec-RNase III cleavage sites identified, around 40% are singletons, in the sense that there is no obvious staggered second cleavage site. Hence, this suggests that they either represent single-stranded cleavages or arise from intermolecular interactions (i.e., the second single-strand cleavage is located in a second molecule thus a complex analysis is required to predict candidate dsRNA partners) [49]. The plasticity of RNase III binding and cleavage sites in *E. coli,* as illustrated in Figure 3 for the *rnc-era*, *rpsO-pnp*, *arfA* and *proP* mRNAs, may provide an explanation for the abundance of putative RNase III cleavage sites (identified by transcriptomic approaches but not yet validated as direct targets) and is consistent with a larger role of RNase III in the regulation of gene expression.

## 3. RNase III Are Everywhere

RNase III enzymes are widely conserved and have been categorized into four classes according to their domain composition (Figure 4). The first one includes all bacterial RNase III (e.g., RNase III and Mini-III) and the yeast RNase III (e.g., Rnt1p and Pac1p) carrying an additional N-terminal domain. The second class includes eukaryotic RNase III carrying additional domains (see Section 3.2). The sole members of class III and IV are the eukaryotic Drosha and Dicer, respectively, where the RIIID is part of complex multidomain proteins. Class I and II RNase III enzymes are directly involved in ribosomal biogenesis and carry a single RIIID per monomer. On the contrary, classes III and IV enzymes carry two RIIIDs and their direct involvement in rRNA maturation has yet to be elucidated. In addition, RNase III enzymes were also found in viruses such as the essential class I RNase III in *Ambystoma tigrinum virus* [70]. Remarkably, RNase III has not been found in archaea where dsRNA cleavage is assured by enzymes belonging to the family of splicing endonucleases [71] which recognize bulge–helix–bulge secondary structure motifs and cut within single-stranded bulges [72].

### 3.1. Bacterial RNase III

The conservation of the RIIID within bacterial genomes allowed the identification of RNase III enzymes in the majority of bacterial species with, so far, the exception of *Deinococcus radiodurans* [6]. Similar to Ec-RNase III [18], RNase III is not essential in most bacteria (e.g., *S. aureus* [73], *C. jejuni* [69], *Borrelia burgdorferi* [74] or *Synechococcus* sp. strain PCC 7002 [55]). However, RNase III was shown to be essential in *B. subtilis* due to its requirement for toxin silencing [75].

To obtain a better understanding of RNase III binding sites and cleavage determinants among species, complementation and substrate specificity assays have often been used. *B. subtilis* RNase III (referred hereafter as Bs-RNase III) exhibits 36% sequence identity with Ec-RNase III and is able to complement the maturation of rRNAs when expressed in an *E. coli rnc* mutant [76]. However, although Bs-RNase III can cleave at the same location of some Ec-RNase III substrates in vitro, the contrary is not valid for the Bs-RNase III targets tested [77]. Of note, *B. subtilis* also contains a shorter form of RNase III that lacks the dsRBD, named Mini-III (hereafter referred to as Bs-Mini-III) which catalyzes the 23S rRNA maturation (see Section 4.1) [78]. Furthermore, while RNase III from *Rhodobacter capsulatus* can cleave some of Ec-RNase III substrates at the exact position in vitro, the contrary is not true, as Ec-RNase III is unable to process the *R. capsulatus* pre-23S rRNA [79,80]. In the cyanobacteria *Synechococcus* sp. strain PCC 7002, three RNase III enzymes were identified, of which one is a homologue to the Bs-Mini-III [55]. Two of them are involved in independent maturation events of the pre-23S rRNA while another participates in plasmid copy number regulation. Interestingly, these enzymes were able to cleave some Ec-RNase III targets in vitro at the same location and other targets at slightly different locations. Thus, a large-scale study of RNase III target selection determinants comparing enzymes and substrates from different species could reveal insights on target selection.

### 3.2. Eukaryotic RNase III

The first eukaryotic RNase III enzymes were identified by sequence comparison with Ec-RNase III. Pairwise comparison of the entire Ec-RNase III revealed 24% identity with Pac1p from the yeast *Schizosaccharomyces pombe* [81,82] and 25% identity with Rnt1p from the yeast *Saccharomyces cerevisiae* [83] (as compared to 36% identity between Ec and Bs-RNase III). Analogous to the known function of bacterial RNase III, Pac1p and Rnt1p were also shown to be involved in rRNA maturation [84,85]. Other RNase III members were identified, thanks to their RIIID signature domain, in higher eukaryotes. Drosha [86] and Dicer [87] are involved in different steps of the maturation of micro RNAs (miRNAs) and silencing RNAs (siRNAs) within the RNA interference pathway [88]. Additional eukaryotic RNase III enzymes demonstrating different domain compositions are represented in Figure 4. They include KREN1 to 3 and mRPN1 from *Trypanosoma brucei* mitochondrial kinetoplast, which contain a C2H2 Zinc finger and whose precise roles remain unclear [89,90], RNC1 in *Zea mays* chloroplasts, whose two RIIIDs are catalytically inactive [91], AtRTL1/2 class II RNase III-like enzymes in *Arabidopsis thaliana* nucleus [92,93] and RNC3/4 mini-RNase III-like enzymes in *Arabidopsis thaliana* chloroplast [94].

#### 3.2.1. Rnt1p and Pac1p

Rnt1p is not essential for viability but, like in *E. coli* [19], its inactivation leads to a severe reduction of the growth rate and temperature sensitivity in *S. cerevisiae* [95]. Pac1p was initially identified to be critical for conjugation and sporulation in *S. pombe* [10,81]. Rnt1p and Pac1p are involved in both ribosomal biogenesis and spliceosomal RNA maturation [83,84]. Rnt1p and Pac1p carry one RIIID and one dsRBD but display a longer N-terminal fragment, relative to prokaryotic RNase III (Figure 4), which was shown to promote the stabilization of the homodimer Rnt1p [96]. Remarkably, Pac1p is able to cleave an Ec-RNase III substrate at the canonical site, while Ec-RNase III fails to cleave Pac1p substrates in vitro [10]. On the contrary to bacterial RNase III, Rnt1p target specificity depends on the interaction of an extended structure of the first helix of the dsRBD with an AGNN tetraloop motif following a minimal 13 bp stem [97]. This initial interaction leads to a structural rearrangement of the dsRBD and tetraloop, allowing the formation of the catalytic site [98]. It is noteworthy that loops of bacterial dsRNA targets are much less conserved (i.e., from four to thousands of nts in the case of the 23S pre-rRNA (see Section 4.1) which can be correlated with the less stringent structural requirement of their dsRBD.

#### 3.2.2. Drosha

The class III RNase III Drosha was first identified in the eukaryotic model *Drosophila melanogaster* [86]. It is a nuclear enzyme initiating pre-miRNA processing from long dsRNA for the RNA interference pathway. Drosha is well conserved in animals and contains two RIIIDs and one dsRBD (Figure 4). In addition, Drosha N-terminus contains a proline-rich region and an arginine/serine (RS)-rich domain responsible for the targeting of pri-miRNAs in the nucleus. Stable binding of dsRNA target by Drosha requires its interaction with the RBP, DiGeorge Syndrome Critical Region 8 (DGCR8) within the microprocessor complex [99]. The N-terminal proline and RS-rich domains were also shown to be involved in the nuclear localization of Drosha [100]. In addition to its role in the interference pathway, Drosha displays various functions including regulation of gene expression, splicing, antiviral defence and may be involved in rRNA maturation (see Section 4.2). Moreover, Drosha mutants are associated with numerous diseases in humans [101].

#### 3.2.3. Dicer

Found in animals, plants and yeast, Dicer is a cytoplasmic enzyme, which generates the mature miRNAs from pre-miRNAs, following the cleavage by Drosha in the nucleus. Dicer is the largest RNase III enzyme (Figure 4). It carries two RIIIDs and multiple additional domains including a DExD/H helicase domain involved in the binding to the terminal loop of targeted pre-miRNAs [102,103,104]. An uncharacterized DUF283 domain was recently shown to contribute to RNA-RNA annealing in human Dicer [105]. The platform (PF) domain serves as a structural measurement for the size of the cleavage product [106] and a PAZ domain anchors the 3′ 2-nts overhanging end of the dsRNA substrate [107]. Hence, Dicer binding and cleavage of a dsRNA target are tightly controlled by both the helicase and the PAZ domains [108]. Interestingly, the dsRBD of Dicer is not essential for target binding but can functionally compensate for the deletion of the PAZ domain, suggesting the involvement of the dsRBD in the interaction with alternative substrates [103]. Dicer plays an essential role in the maturation of miRNA from pre-miRNA, which are then loaded on the RNA-induced silencing complex (RISC) together with an Argonaute protein. Moreover, Dicer is also involved in the maturation of other regulatory RNAs (e.g., siRNAs and 5′ tRNA fragments) and may be involved in rRNA maturation (see Section 4.2) [109]. In regard to the many roles of Dicer, it is not surprising that its inactivation greatly impacts cell viability and provokes premature death in mice [110]. Interestingly, mutants of Ec-RNase III (E38A and E65A/Q165A) were found to be able to produce 23 nts fragments from long dsRNA stems in a similar way to the Dicer enzyme in the process of siRNA maturation [22,111].

## 4. Maturation of rRNA by RNase III

### 4.1. Bacteria

#### 4.1.1. The Case of *E. coli*

In bacteria, the large subunit (50S) contains the 23S and 5S rRNAs and 33 proteins while the small subunit (30S) contains the 16S rRNA and 21 proteins [112]. In *E. coli*, seven rRNA operons are responsible for the co-transcription of the 16S, 23S and 5S rRNA as well as several interspaced tRNAs, the identity of which depends on the rRNA operon. The maturation of pre-rRNA relies on an initial series of RNase III processing events [12,113,114,115]. RNase III binds to two dsRNA duplexes flanking the 16S and 23S rRNAs (Figure 5) [116]. Hence, RNase III generates the 17S precursor containing +115 and +33 nts from the mature 5′ and 3′ ends respectively of the 16S rRNA [115,117]. Maturation of the pre-23S rRNA is more complex as RNase III cleavage release 23S pre-rRNAs containing +3, +7 or −4 nts from the mature 5′ end and +7 nts from the 3′ end (see Section 5.3) [115]. This later cleavage releases the 9S precursor containing +85 nts from the 5S rRNA to the terminator of the polycistronic transcript. Moreover, RNase III processing occurs co-transcriptionally, allowing the uncoupling of transcription and further steps of maturation (Figure 5A) [118]. In vivo evidence suggests that the formation of the RNase III cleavage site within the 16S rRNA is aided by a protein chaperone complex composed of the RNA polymerase-associated factors NusA and NusB, consistent with the co-transcriptional activity of RNase III [119]. RNase III processed pre-rRNAs are subsequently cleaved by endoribonucleases (e.g., RNase E, G and unknown) and trimmed by exoribonucleases (e.g., RNases R, II, PH and PNPase) at their 3′ ends to generate mature rRNAs [2].

#### 4.1.2. Alternative Cleavages

Despite its involvement in the uncoupling of transcription and maturation, RNase III is not essential to produce functional rRNAs since processing by other RNases can still occur in an *rnc* mutant [114,120]. However, the accumulation of full-length pre-rRNAs in *rnc* mutants suggests that the alternative maturation pathway is slower than the transcription of the whole operon [121]. Furthermore, in the absence of RNase III, pre-23S RNAs are not completely matured (yielding species up to 100 nts longer at both 5′ and 3′ ends) but are still assembled into functional 50S subunits [114]. RNase III processing of the pre-23S rRNA occurs +3 or +7 nts from the 5′ end and +7 nts from the 3′ end of the mature 23S rRNA. Despite these observations, the precise role of RNase III in the maturation of the pre-23S rRNA is not totally elucidated. First, contrary to the expected dsRNA cleavage of the 23S rRNA, the 5′ and 3′ cleavages of one 23S pre-rRNA were shown to occur independently [122]. Second, RNase III can associate with 30S, 50S and 70S ribosomes in vitro and third, pre-23S rRNA maturation by RNase III can be different when pre-23S rRNA are assembled within ribosomal subunits (in the absence of an initial RNase III cleavage event) [123]. Indeed, RNase III cleaves in vitro the pre-23S rRNA 3 nts upstream from the mature 5′ extremity of the 23S rRNA within 70S ribosomes while both pre-23S rRNA extracted from an *rnc* mutant and in vitro synthesized 23S rRNA are cleaved by RNase III 7 nts upstream of the 5′ end of the mature 23S rRNA. In addition, RNase III is responsible for the maturation of a shorter 23S pre-rRNA in which the 5′ terminus contains 4 nts less than the main 23S form [115]. This shorter 23S rRNA was observed to accumulate during the stationary phase of growth but its relevance remains unclear. Hence, these observations suggest that RNase III could be involved in alternative maturation paths of the pre-23S rRNA according to its assembly state in the ribosome and under specific growth conditions (e.g., when RNase III is inactivated during osmotic stress, cold shock or entry into the stationary phase [124]).

Moreover, *rnc* mutants were recently shown to carry 30S subunits composed of intermediate 17S* rRNA containing 133 nts upstream and 49 nts downstream from the mature 16S rRNA [125]. These subunits are not found in 70S ribosomes suggesting that they are either further processed via an unknown path or cannot be incorporated in functional ribosomes. In the same study, deletion of the gene *rpsO*, encoding the ribosomal protein S15, was shown to lead to the accumulation of heterogeneous pre-16S rRNAs which are incorporated into 70S ribosomes [125]. As S15 inactivation results in cold sensitivity and cell elongation, the subsequent inactivation of RNase III can partially restore growth and a normal cell length. In the *rnc*/*rpsO* double mutant, 70S ribosomes containing mature 16S rRNA are more abundant than in the *rpsO* mutant at 25 °C. Hence, the authors suggest that the 16S rRNA could be matured via an alternative pathway in the absence of RNase III, which may be associated with slower pre-rRNA maturation.

A very clear example of a ribosomal protein affecting RNase III cleavage was demonstrated for Bs-Mini-III, whose activity depends upon the presence of the ribosomal protein L3 in *B. subtilis* [126]. Recently, ribosome-bound Bs-Mini-III intermediate structures obtained by crystallography shed light on the process by which the pre-23S rRNA is matured within assembled 50S large subunits [127]. Hence, Mini-III was suggested to participate in the final quality control step for *B. subtilis* ribosome biogenesis.

#### 4.1.3. Conservation

The role of RNase III in the initial steps of rRNA processing is conserved among a large range of bacterial species including but not limited to *B. subtilis* [76], *B. burgdefori* [128], *A. aeolicus* [129], *T. maritima* [130], *P. putida* [66], *Lactococcus lactis* [131] or *Helicobacter pylori* [132]. However, differences that may reflect alternative evolutionary paths have also been observed. In *H. pylori*, the 16S rRNA gene is encoded independently from the 23S-5S rRNA. Both are matured by RNase III as in *E. coli* with the exception of an atypical dsRNA structure located upstream from the 5S rRNA [132]. Remarkably, an additional maturation event was found within the leader region of the 23S-5S rRNA operon, which required the binding of a convergently transcribed asRNA [133]. The physiological role of this asRNA-driven maturation event remains unclear. It was suggested that it may play a role in the recycling of the processed 5′ fragment and support the step-by-step folding of the pre-rRNA. The 16S gene from *B. burgdorferi* is also isolated from the 23S-5S rRNA operon [128]. However, although *B. burgdorferi* 23S and 5S rRNA are canonically matured by RNase III, the formation of 16S rRNA was not affected in an *rnc* mutant.

Furthermore, specific ribosomal sequences termed intervening sequences (IVS) have been identified in some 23S rRNA genes in various bacterial species (e.g., *S. enterica*, *R. sphaeroides*, *Campylobacter* sp. but not in *E. coli*) [134,135,136]. IVS are 100–200 nts sequences forming an additional stem in 23S rRNAs which are excised by RNase III and lead to the fragmentation of 23S rRNAs [135]. Interestingly, the presence of the IVS (in an *rnc* mutant) or its removal leaving the rRNA in two pieces has no detected consequences on ribosome function [137]. The precise role of IVS remains unclear; however, evidence in *Yersinia enterocolitica* and *S. typhimurium* suggests that rRNA fragmentation could provide fitness advantages in the eukaryotic host to protect rRNA fragments from bacteriocins [138] and to exit the stationary phase [139].

### 4.2. Eukaryotes

Ribosome structure and complexity greatly increase from prokaryotes to eukaryotes. In yeast, the 60S large subunit contains the 5S, 5.8S and 25S rRNAs, while the 40S small subunit contains the 18S rRNA. Eukaryotic rRNAs are transcribed by RNA polymerase I as a single pre-rRNA, except the 5S rRNA, which is transcribed by RNA polymerase III [140]. In *S. cerevisiae*, Rnt1p participates in the maturation of the 35S pre-rRNA into pre-25S rRNA by cleavage of a dsRNA 14 and 49 nts downstream from the mature 25S 3′ end, in a reaction, which requires the U3 small nucleolar RNA [141,142]. Inactivation of Rnt1p leads to the accumulation of the 35S pre-rRNA in vivo but, similar to *E. coli*, alternative processing events can partially maintain ribosome biogenesis [84]. The RNase III enzyme Pac1p of *S. pombe* demonstrates a similar involvement in rRNA maturation with the difference that cleavage occurs 41 and 83 nts downstream from the mature 25S 3′ end [143].

In eukaryotes, rRNA maturation relies on the complex interplay of numerous proteins and regulatory actors from the nucleus to the cytoplasm [140]. The inactivation of the nuclear Drosha and the cytoplasmic Dicer in humans revealed an accumulation of the 12S pre-rRNA, suggesting that both enzymes are involved in the maturation of the 5.8S rRNA [9,144]. However, this function has not been validated directly. Furthermore, because the level of mature rRNAs is not significantly affected in cells inactivated for either Drosha or Dicer, their involvement in ribosome biogenesis remains unclear.

In the plant *Arabidopsis thaliana*, the class II RNase III AtRTL2 plays a direct role in the maturation of the 3′ end of the pre-23S rRNA in the nucleus [92]. In addition, the two chloroplastic mini-RNase III-like enzymes, RNC3 and RNC4 (75% identity with Bs-Mini-III) are involved in rRNA maturation [94]. Inactivation of both RNC3 and RNC4 lead to a strong reduction in chlorophyll accumulation in leaves, which is partially due to their role in the maturation of the 5′ end of the 23S rRNA and the 3′ end of the 4.5S rRNA. It is also interesting to note that RNC3 and RNC4 were proposed to be involved in alternative rRNA processing events dependent upon two asRNAs, complementary to the 5S rRNA and 4.5S-5S rRNAs, which is reminiscent of the asRNA overlapping the pre-23S-5S rRNA in *H. pylori* [132,145].

## 5. Additional Roles of RNase III in Ribosome Biogenesis

### 5.1. Indirect Regulation of Ribosome Biogenesis

In bacteria, RNase III was shown to play an important role in the regulation of gene expression by processing mRNAs and regulatory RNAs. Ec-RNase III plays an accessory role in the expression of ribosome biogenesis factors. For example, the essential GTPase Era, co-expressed with and repressed by RNase III [37,146], coordinates the folding of the 30S subunits by binding within the anti-Shine-Dalgarno sequence in the 3′ end of the 16S rRNA [147,148]. It is noteworthy that the Era binding site on the 16S rRNA is located close to the RNase III binding site. However, they act at different steps of rRNA maturation and do not interact [149]. In addition, Era binding was suggested to act as a checkpoint for the initiation of cell division [150] and was later shown to also inhibit cell division upon overexpression [35] implying that RNase III could have an indirect role in coordinating protein synthesis with cell division. This hypothesis is further supported by the observation that *rnc* inactivation leads to a reduction in cell length [125].

Similarly, RNase III cleaves the *rpsO-pnp* polycistronic mRNA, leading to the destabilization of the *pnp* mRNA (Figure 3), encoding the exoribonuclease PNPase essential at low temperature [151,152,153]. Consequently, PNPase accumulates during a cold shock following the induction of YmdB, an inhibitor of Ec-RNase III activity [154]. PNPase was shown to be involved in the maturation of the 3′ end of the 16S rRNA [155]. Hence in the absence of RNase III, the increased abundance of PNPase could compensate for the slower 16S rRNA maturation. It is worth noting that RNase III repression of PNPase expression is conserved in *B. burgdorferi* [74], *S. coelicolor* [156], *S. pyogenes* [157] and *C. jejuni* [158]. RNase III is also involved in the repression of the expression of the transcription factor NusA and the essential translation initiation factor IF2, which is also induced during a cold shock [159]. Hence, during a cold shock, the repression of RNase III activity allows PNPase, NusA and IF2 to accumulate, which is supposedly required to maintain efficient ribosome biogenesis and translation at low temperatures. Supporting this hypothesis, RNase III inactivation was shown to partially rescue growth at low temperature in strains inactivated for *nusA*, *nusB* or *rpsO* [119,125].

In a recent RNA-seq study, thousands Ec-RNase III in vivo cleavage sites were identified [49]. Cleavage sites were found in mRNAs encoding ribosomal proteins, translation elongation factors and 23% (18 out of 80) of the “RNA modifiers” (i.e., enzymes involved in RNA modifications) including the 23S methyltransferase and the 23S pseudouridine synthase. Although these RNase III cleavages have yet to be characterized individually for effects on ribosome biogenesis, the importance of Ec-RNase III in the regulation of ribosomal biogenesis factors is likely to be larger than previously suspected.

In eukaryotes, RNase III enzymes are essential for the maturation of non-coding RNAs of the interference pathway. In the yeast *S. cerevisiae*, Rnt1p matures small nucleolar RNAs, snR39 and snR59, from within introns of the RPL7A and RPL7B ribosomal protein-encoding mRNAs [160]. The processing of snR39 and snR59 prevents the splicing of the mature RPL7A and RPL7B mRNAs, suggesting that Rnt1p can control the expression of ribosomal proteins RPL7A and RPL7B. However, because of the highly pleiotropic role of non-coding RNAs in eukaryotes, the importance of indirect regulation of ribosomal factors is still unclear.

### 5.2. Antibiotic Resistance

The characterization of an *E. coli* isolate resistant to sublethal concentrations of aminoglycosides antibiotics (i.e., kanamycin and streptomycin) was associated with an increased RNase III activity [64]. In this strain, the increased RNase III activity leads to reduced expression of *rng*, encoding RNase G, a direct target of RNase III. Reduction of RNase G expression provokes the accumulation of 16.3S rRNA in which 6 to 8 nts are not processed from the 5′ end. Significantly, ribosomes containing 16.3S rRNAs are partially resistant to aminoglycosides. Supporting this observation, it was later shown that the overexpression of YmdB, which reduces the activity of RNase III, led to increased sensitivity to aminoglycoside antibiotics (i.e., apramycin and gentamicin) while deletion of *ymdB* increased resistance to aminoglycoside antibiotics (i.e., apramycin, gentamicin, streptomycin and neomycin) [161]. Tetracycline antibiotics also inhibit translation by binding to ribosomal subunits. The tetracycline antibiotics, doxycycline and minocycline, were shown to bind 27 nts dsRNAs in vitro and to prevent RNase III cleavage activity on specific substrates [162]. Another report suggested that treatment of *E. coli* with doxycycline can lead to the accumulation of pre-rRNAs, hence the authors hypothesized that the binding of the drug could prevent the initial maturation of pre-rRNA by RNase III [163].

### 5.3. Intracellular Localization

Transcriptomic analysis and in situ microscopy imaging have revealed the unexpected localization of specific RNAs and proteins in cells (e.g., accumulation of sRNAs and the RBP chaperone Hfq at the poles under osmotic stress [164]). RNase III was previously shown to be enriched within inner membrane fractions [165]. Contradictory results were obtained by the in situ imaging of RNase III, which revealed a homogeneous repartition in *E. coli* [166]. On the other hand, in situ RNA imaging revealed the colocalization of the 5′ end of pre-rRNA with the nucleoid (unlike mature ribosomes), which were equally distributed in the cytoplasm [166]. Significantly, the nucleoid localization of the pre-rRNA was lost in two independent *rnc* mutants (*rnc14* and *rnc70*, Table 1), suggesting that RNase III could participate (directly or indirectly) in the localization of pre-rRNAs in the nucleoid during their initial maturation step by RNase III. Another indication of the role of RNase III in nucleoid function is the observation that the inactivation of the nucleoid-associated protein Fis is lethal in an *rnc* mutant [166]. In addition, the yeast Rnt1p interacts with the RNA polymerase I in the nucleus and was co-immunoprecipitated with ribosomal DNA suggesting that Rnt1p is involved in both maturation and transcription of rRNAs [167]. Together these observations indicate that there are additional roles of RNase III related to ribosome biogenesis to be explored.

## 6. Conclusions

RNase III are versatile enzymes demonstrating both common and unique characteristics in bacteria and eukaryotes. In bacteria, RNase III can bind and recognize a wide range of RNA targets and is involved in the regulation of ribosome biogenesis at multiple levels. While class-II eukaryotic RNase III enzymes also participate in the direct maturation of pre-rRNAs, the involvement of Class III and IV enzymes Drosha and Dicer in ribosome assembly requires clarification. Furthermore, as demonstrated in *E. coli* and other bacteria, RNase III is not only involved in the initial steps of pre-rRNA maturation but also in the regulation of the expression of multiple ribosome biogenesis factors. Hence, elucidating the role of RNase III in the step-by-step maturation and assembly of the ribosome could reveal insights into alternative ribosome biogenesis pathways as a key to understanding ribosome heterogeneity.

## Figures and Tables

**Figure 1 microorganisms-09-02608-f001:**
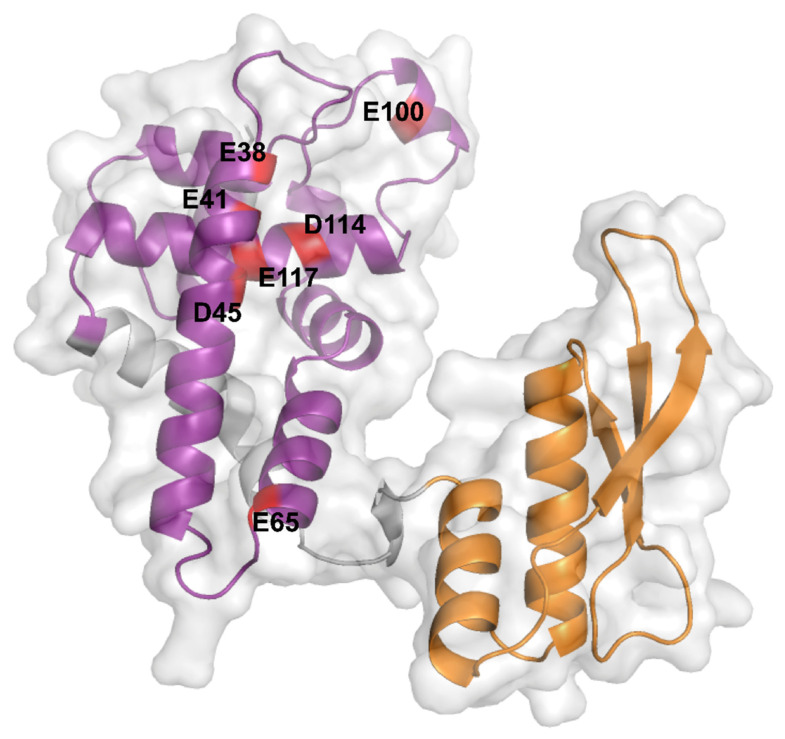
AlphaFold prediction of the Ec-RNase III structure. Ec-RNase III structure predicted by the AlphaFold program (AF-P0A7Y0-F1, https://alphafold.ebi.ac.uk, accessed on the 11 October 2021). The N-terminal RIIID (6-128) and C-terminal dsRBD (155-225) are depicted in purple and orange, respectively. Critical negatively charged residues E38, E41, D45, E65, E100, D114 and E117, which are highly conserved among bacterial RNase III enzymes, are highlighted in red on the structure of the Ec-RNase III. The representation was generated using the PyMol software version 2.5.1.

**Figure 2 microorganisms-09-02608-f002:**
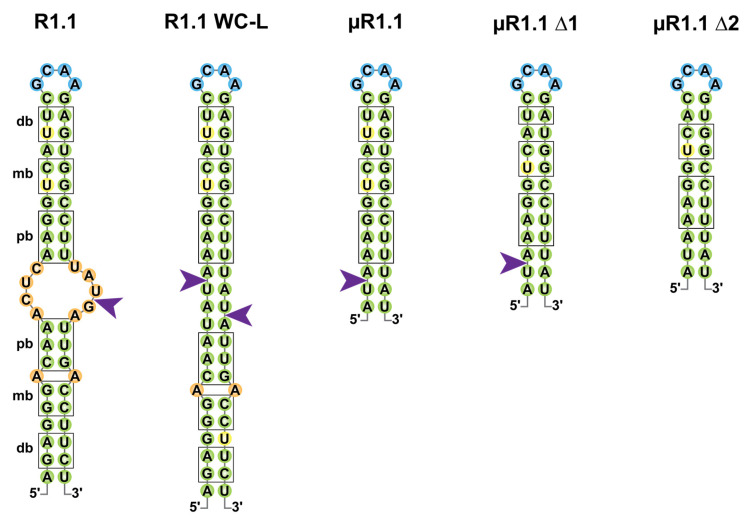
Cleavage of the T7 R1.1 RNA and its derivatives by Ec-RNase III. Secondary structures of RNase III target R1.1 RNA from the T7 phage and its artificial derivatives (R1.1 WC-L, µR1.1, µR1.1 Δ1 and µR1.1 Δ2) are represented in two dimensions as in [29,48]. Base pairs are represented in green; bulges and mismatches are represented in orange, loops in blue and uridine residues involved in wobble G-U base pairing in yellow. Proximal (pb), middle (mb) and distal (db) boxes are shown outlined in black and RNase III single-strand cleavage sites are represented by a purple arrow.

**Figure 3 microorganisms-09-02608-f003:**
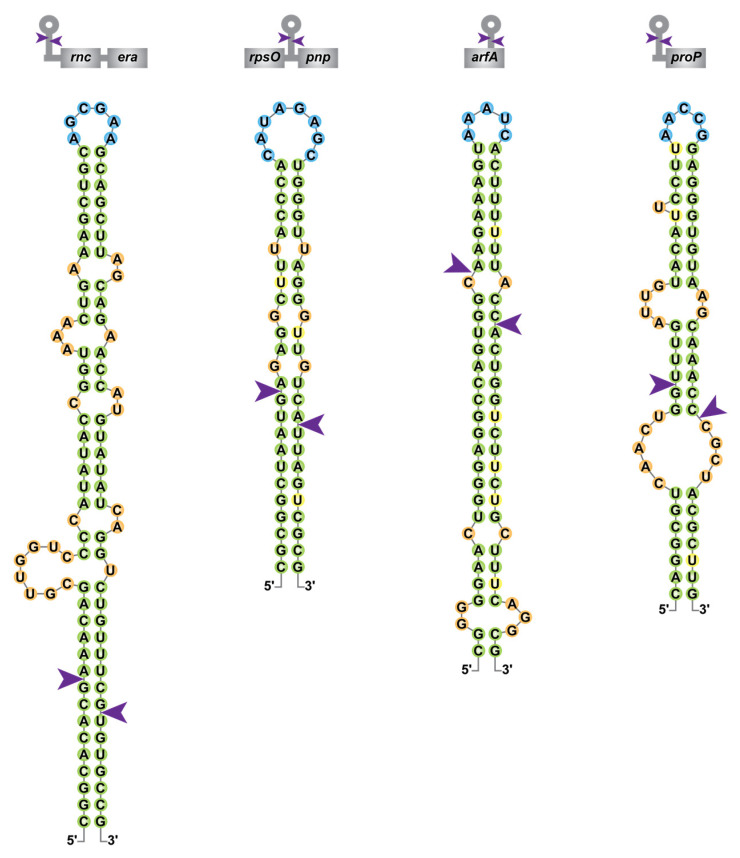
Diversity of RNase III cleavage sites within *E. coli* mRNAs. Secondary structure predictions of Ec-RNase III targets within the *rnc*-*era*, *rpsO*-*pnp*, *arfA* and *proP* mRNAs were obtained from [49,55,62,70] and color-coded as in Figure 2. A schematic representation of the targeted mRNAs is presented on top of each RNA structure with coding sequences in grey boxes.

**Figure 4 microorganisms-09-02608-f004:**
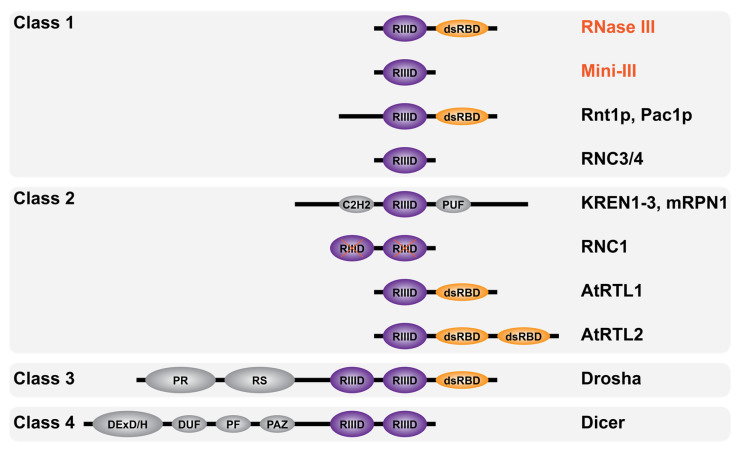
Domain diversity of RNase III enzymes. Schematic representation of RNase III enzymes domain composition categorized by classes as described in the text (not to scale). Bacterial RNase III are in red font and eukaryotic RNase III in black font. The RNase III catalytic domain (RIIID) is in purple and marked with a red X when inactive (in RNC1), the double-stranded RNA binding domain (dsRBD) is in orange while the Zinc finger domain C2H2, RNA-binding domain PUF, proline-rich domain PR, arginine/serine (RS)-rich domain, helicase DExD/H domain, RNA annealing domain DUF, structural domain PF and the anchoring domain PAZ are in gray.

**Figure 5 microorganisms-09-02608-f005:**
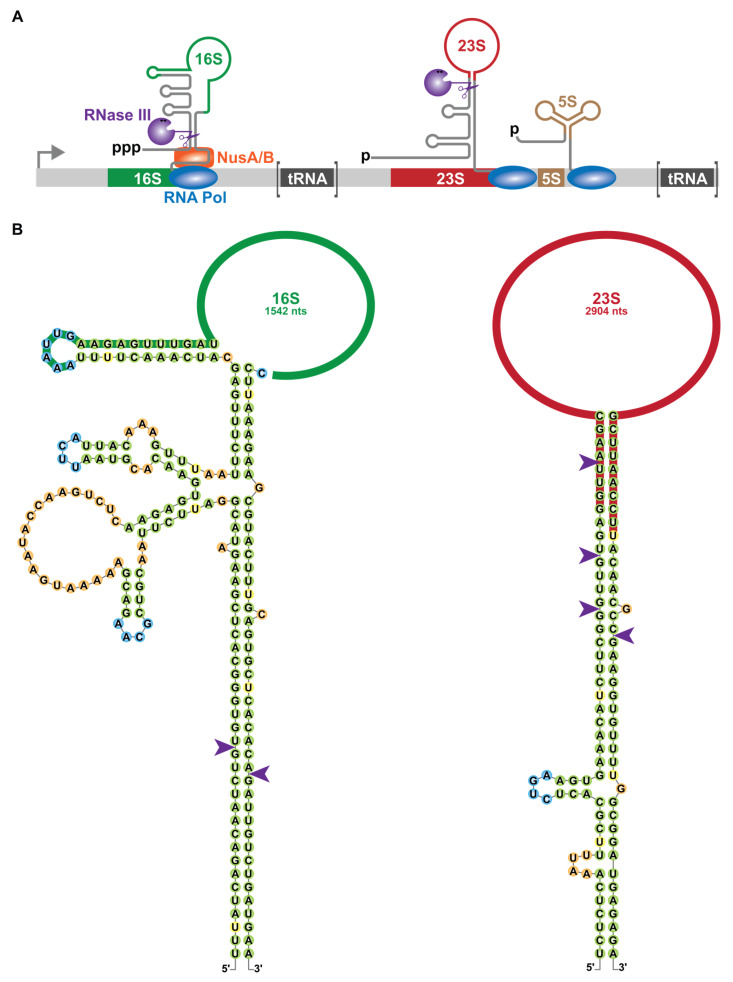
Co-transcriptional maturation of rRNAs by RNase III in *E. coli*. (**A**). Schematic representation of an rRNA operon (not to scale). RNase III (purple) initiates the processing of the three rRNAs (16S in green, 23S in red and 5S in brown in a prototypical ribosomal operon) during transcription by RNA polymerase (blue) starting from the indicated promoter (bent arrow). NusA and NusB chaperone proteins are represented in orange, 5′ extremity of the triphosphorylated rRNA is represented by “ppp” and 5′ monophosphate extremities of the pre-23S and pre-5S rRNAs generated by RNase III cleavages are represented by “p”. (**B**). Secondary structures of Ec-RNase III cleavage sites within the 16S (in green) and 23S (in red) rRNAs of the *rrsH*-*rrlH*-*rrfH* operon were obtained from [116] and color-coded as in Figure 2.

**Table 1 microorganisms-09-02608-t001:** Functionally characterized RNase III mutants in *E. coli*.

Allele	Mutation	Catalytic Activity		In Vitro dsRNA Binding	References
		In Vivo	In Vitro		
	E38A	n.d.	weak ^1^	high	[21,22]
	E38V	no	n.d.	n.d.	[23]
	E41A	n.d.	weak ^1^	low	[21]
	F42G, D or R	no	n.d.	n.d.	[23]
	F42M, W	yes	n.d.	yes	[23]
*rnc105*	G44D	no	no	no	[24,25]
	D45A	no	low	yes	[21,23]
	D45E or N	n.d.	weak ^1^	low	[21]
	E65A	no	weak ^1^	low	[21,23]
*rnc97*	G97E	low	weak ^1^	n.d.	[26]
	E100A	n.d.	weak ^1^	low	[21]
	D114A	n.d.	weak ^1^	low	[21]
	E117D	n.d.	weak ^1^	yes	[23,27]
*rnc70*	E117K, A or Q	no	no	yes	[28,29,30]
*rnc10*	Q153P	n.d. ^2^	weak	n.d.	[31]
*rnc7*	D155E	n.d. ^2^	no	n.d.	[31]
*rev*3	A211V	n.d. ^2^	n.d.	n.d.	[25,32]
*rnc38*	insertion	no	no	no	[33]
*rnc40*	insertion	no	no	no	[18]
*rnc14*	insertion	no	no	no	[18]

Mutations are either missense, indicated by the amino acid, its number relative to the sequence of Ec-RNase III followed by the replacing amino acid(s) or insertions of fragments derived from transposons and plasmids. ^1^ In excess of magnesium. ^2^ Suppressors of cold-sensitive mutant alleles. n.d. not determined.

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
