# Peer review of "RNase III, Ribosome Biogenesis and Beyond"

_microorganisms, 2021, doi:10.3390/microorganisms9122608_

Round 1

Reviewer 1 Report

The manuscript entitled “RNase III, ribosome biogenesis and beyond” by Lejars, Kobayashi, and Hajnsdorf reviewed previous studies regarding RNase III. This review is comprehensive ranging from structure to function in bacteria and eukaryote and clearly written in most of the parts. I feel that this manuscript would become useful literature to grasp the current understanding for RNase III. I recommend publication in the current form after clarifying the following points.

  1. P2, lane 61-65. The authors probably indicate a mistake in the Ecocyc database. I recommend the authors inform the responsible person in the Ecocyc database about the mistake instead of writing in a journal paper because if the database corrected the mistake in the future, the description in this review would be incomprehensible.

  1. P3, lane84-85. “These mutations are located either in the RIIID or in the dsRBD(Figure 1).” Please indicate which mutations are in the dsRBD or indicate the amino acids numbers for RIIID and dsRBD.

  1. Figure 1. Why did authors use AlphaFold prediction, instead of experimentally analyzed structure? If these structures are almost identical, please explain in the legend.

  1. P5, lane 174-179. The sentence that starts from “RNase III is also involved in t intermolecular cleavages of…” is unclear. Does the phrase, “intermolecular cleavages”, mean the cleavage of intermolecular dsRNA?

  1. P9, lane 326. “Figure 4” is probably a mistake of Figure 5.

  1. P13, lane 482-485. The sentence that starts from “Second, as it was shown in vitro that …” is unclear. Please rephrase.

Author Response

 We would like to thank the Reviewers for their constructive criticisms and helpful suggestions for improving the manuscript. All references to pages and line numbers refer to the initially submitted version of the manuscript, the major modifications are in red and deletions in the text are not indicated. Due to reorganization of the text, reference numbering in the revised manuscript has been changed.

P2, lane 61-65. The authors probably indicate a mistake in the Ecocyc database. I recommend the authors inform the responsible person in the Ecocyc database about the mistake instead of writing in a journal paper because if the database corrected the mistake in the future, the description in this review would be incomprehensible.

The note was removed from the manuscript and a report error was submitted on the Ecocyc database. Ingrid Keseler confirmed the error and the correction will be effective in the next update of the Ecocyc database (Spring 2022).

    P3, lane84-85. “These mutations are located either in the RIIID or in the dsRBD(Figure 1).” Please indicate which mutations are in the dsRBD or indicate the amino acids numbers for RIIID and dsRBD.

The sentence was corrected to indicate that RNase III mutant in which catalytic activity is disrupted include Q153 in the linker and D155E in the dsRBD.

    Figure 1. Why did authors use AlphaFold prediction, instead of experimentally analyzed structure? If these structures are almost identical, please explain in the legend.

The section 2.2.1 was corrected to highlight the fact that no complete structure of E. coli RNase III is available to date. Hence, we mapped the characterized E. coli RNase III mutations to the AlphaFold predicted structure of Ec-RNase III.

    P5, lane 174-179. The sentence that starts from “RNase III is also involved in t intermolecular cleavages of…” is unclear. Does the phrase, “intermolecular cleavages”, mean the cleavage of intermolecular dsRNA?

The sentence was clarified.

    P9, lane 326. “Figure 4” is probably a mistake of Figure 5.

The numbering was corrected.

    P13, lane 482-485. The sentence that starts from “Second, as it was shown in vitro that …” is unclear. Please rephrase.

The sentence was corrected as also recommended by Referee 2.

Reviewer 2 Report

This is an interesting review focusing on the diversity of RNase III enzymes and their roles in ribosome biogenesis. The authors describe the structure and mechanism of action of bacterial RNase III, the diversity of its substrates and its involvement in diverse processes, and the different eukaryotic RNase III enzymes. Then, the role of RNase III in rRNA maturation in both bacteria and eukaryotes is discussed. In the last part of the manuscript, additional roles of RNase III in ribosome biogenesis are considered.

The numerous bibliographic references are generally well cited and recent studies in the field are included. However, some points need to be clarified (see major and minor comments).

Major comments:

1) Section 5.3: Additional roles of RNase III in ribosome biogenesis / rRNA maturation pathways.

In this chapter, the authors describe some well-characterized RNase III processing cleavages of pre-23S rRNA (lines 477-495), an alternative 16S rRNA maturation pathway observed in absence of RNase III (lines 496-504) and finally data on the final maturation step of 23S rRNA maturation by Mini-III in B. subtilis (lines 505-510). I do not understand why these examples of 23S and 16S rRNA processing by RNase III and Mini-III are reported in a separate chapter on “Additional roles of RNase III in ribosome biogenesis” and not in the chapter 4 (“Maturation of rRNA by RNase III in bacteria”). Thus, I suggest moving this chapter to section 4.1 to avoid redundancy (particularly for 23S rRNA maturation) and to allow an easier reading.

2) Figure 4: The authors should explain what is the basis for their classification of RNase III enzymes. In contrast to their classification, bacterial RNase III and yeast Rnt1/Pac1 have been grouped in class 1 in several other studies. The authors should change “ArTL1” and “ArTL2” by AtRTL1 and AtRTL2, respectively. It would also be useful to precise to which organisms belong the different RNase III enzymes.

Minor comments:

L 34: The authors should distinguish between RNA-binding proteins and RNA chaperones: Whereas specific RNA-binding proteins have an RNA chaperone activity (they help RNAs to fold), not all RBPs are chaperones. Replace chaperon by chaperone.

L40: The reference 12 is not appropriate here: In this article, purified RNase III has been shown to digest double stranded polyribonucleotides, not rRNA.

L54-56: The citation of Kindler et al (ref14) is confusing here. It seems that Kindler et al isolated RNase III mutants from a mutagenized library made by (16). This is not quite correct. Indeed, in ref 16, two E. coli RNase I mutants were isolated whereas in ref 14, they mutagenized the RNase I mutant to screen for RNase III mutants.

L57-58: If I am correct, the rnc105 mutant was tested only in vitro in ref 14.

L75-76: This sentence is not clear: what the authors mean by “requires compensating the expression of downstream genes”. Do the authors mean that the deletion of rnc would lead to a polar effect leading to inactivation of era expression?

L116: ref 41 and 42 should be moved just after (https://alphafold…)

L121: I would replace “Function” by “mechanism”.

L147: Replace µR1.1∆2 by µR1.1∆1

P160: To be more precise, I would replace “Physiology” by a more precise title such as ”Functional roles of RNase III”.

L161-164: It would be interesting to know what conclusion can be drawn from the different transcriptomics studies. For instance, have these studies identified common targets or common regulatory mechanisms in the different organisms?

L165-173: Rather than enumerating the RNase III cleavages in different mRNAs, it would be more interesting to know (when possible) the consequences of these cleavages on the expression of the corresponding proteins. If I am correct, RNase III has been shown to control the expression of the ArfA protein providing an interesting mechanism to induce an alternative ribosome rescue pathway.

L186-187: The authors state that RNase III is important for functions that are essential for virulence in different bacterial pathogens. They should replace “essential for virulence” by “involved in virulence”. In the cited references, there is no evidence for a role of RNase III in virulence.

L188-194: The authors should clarify why “the second single-strand cleavage …cannot be observed by classic transcriptomic approaches”. In this study (ref 49), the authors attempted to identify trans sense-antisense duplexes, although their number was low.

L194-197: This sentence is not clear. What the authors mean by “explanation for the abundance of unsuspected RNase III cleavage sites”? Do the authors mean that transcriptomics data cannot detect all RNase III cleavage sites?

L271: The authors could verify whether reference 84 is appropriate here.

L289: Replace “DiGeroge” by “DiGeorge”.

L325: This sentence is not correct. Indeed, RNase III binds double-stranded regions flanking the 16S (and not 16S/23S) and 23S (not 23S/5S) mature rRNAs.

L326-327: Precise that the additional 115 nt and 33 nt of the precursor are at the 5’ and 3’ ends respectively.

L331: Replace “decoupling” by “uncoupling”

L351: Reference 116 would be more appropriate than ref 122.

L404: The “class I ArTL2” belongs to class II in Figure 4.

L450-452: This sentence is vague. What is the link between RNA interference and ribosome biogenesis?

L473-475: The authors might be more cautious with the interpretation of the data presented in reference 161. Indeed, this study has some shortcomings (the quality of the gels and Northern experiments is not satisfactory and the pre-rRNA have not been characterized) making it difficult to draw any conclusion.

L489-495: The authors indicate that RNase III produces a pre-23S rRNA four nucleotides shorter at the 5’ terminus than mature 23S rRNA suggesting that RNase III could be involved in alternative maturations pathways. However, this species has been found in both wild-type and RNase III deficient strain (ref 117) and the significance of this unusual 5’ terminus is not clear (see review of Srivastava and Schlessinger Annu. Rev. Microbiol. 1990).

Author Response

We would like to thank the Reviewers for their constructive criticisms and helpful suggestions for improving the manuscript. All references to pages and line numbers refer to the initially submitted version of the manuscript, the major modifications are in red and deletions in the text are not indicated. Due to reorganization of the text, reference numbering in the revised manuscript has been changed.

1) Section 5.3: Additional roles of RNase III in ribosome biogenesis / rRNA maturation pathways.

In this chapter, the authors describe some well-characterized RNase III processing cleavages of pre-23S rRNA (lines 477-495), an alternative 16S rRNA maturation pathway observed in absence of RNase III (lines 496-504) and finally data on the final maturation step of 23S rRNA maturation by Mini-III in B. subtilis (lines 505-510). I do not understand why these examples of 23S and 16S rRNA processing by RNase III and Mini-III are reported in a separate chapter on “Additional roles of RNase III in ribosome biogenesis” and not in the chapter 4 (“Maturation of rRNA by RNase III in bacteria”). Thus, I suggest moving this chapter to section 4.1 to avoid redundancy (particularly for 23S rRNA maturation) and to allow an easier reading.

As recommended, section 5.3 was moved and integrated in the section 4.1 as suggested. As a consequence, reference numbering has been changed in the revised version of the manuscript.

2) Figure 4: The authors should explain what is the basis for their classification of RNase III enzymes. In contrast to their classification, bacterial RNase III and yeast Rnt1/Pac1 have been grouped in class 1 in several other studies. The authors should change “ArTL1” and “ArTL2” by AtRTL1 and AtRTL2, respectively. It would also be useful to precise to which organisms belong the different RNase III enzymes.

The classification was corrected to follow previous studies, including Rnt1p, Pac1p and RNC3/4 as class I RNase III enzymes. The suggested corrections were introduced in the text and in Figure 4 and in its legend.

Minor comments:

L 34: The authors should distinguish between RNA-binding proteins and RNA chaperones: Whereas specific RNA-binding proteins have an RNA chaperone activity (they help RNAs to fold), not all RBPs are chaperones. Replace chaperon by chaperone.

The sentence was corrected and “chaperon” corrected throughout.

L40: The reference 12 is not appropriate here: In this article, purified RNase III has been shown to digest double stranded polyribonucleotides, not rRNA.

This reference has been replaced by Nikolaev, N.; Schlessinger, D.; Wellauer, P.K. "30 S pre-ribosomal RNA of Escherichia coli and products of cleavage by ribonuclease III: Length and molecular weight." J. Mol. Biol. 1974, 86, 741-748, doi:10.1016/0022-2836(74)90350-7.

L54-56: The citation of Kindler et al (ref14) is confusing here. It seems that Kindler et al isolated RNase III mutants from a mutagenized library made by (16). This is not quite correct. Indeed, in ref 16, two E. coli RNase I mutants were isolated whereas in ref 14, they mutagenized the RNase I mutant to screen for RNase III mutants.

The sentence was corrected as recommended.

L57-58: If I am correct, the rnc105 mutant was tested only in vitro in ref 14.

The sentence was corrected.

L75-76: This sentence is not clear: what the authors mean by “requires compensating the expression of downstream genes”. Do the authors mean that the deletion of rnc would lead to a polar effect leading to inactivation of era expression?

Deletion of rnc in the Keio collection could not be obtained, suggesting polar effects on era expression upon inactivation of RNase III. Yet, insertion mutants (rnc14 and rnc38 carrying large inserts containing IS and fragments of plasmid) were constructed and seem to provide the correct compensation of era expression to be viable. This sentence was clarified.

L116: ref 41 and 42 should be moved just after (https://alphafold…)

The references were moved.

L121: I would replace “Function” by “mechanism”.

The section title was edited as suggested.

L147: Replace µR1.1∆2 by µR1.1∆1

The error was corrected.

P160: To be more precise, I would replace “Physiology” by a more precise title such as ”Functional roles of RNase III”.

The section title was edited as “Physiological roles of RNase III”.

L161-164: It would be interesting to know what conclusion can be drawn from the different transcriptomics studies. For instance, have these studies identified common targets or common regulatory mechanisms in the different organisms?

Global approaches aimed at identifying RNase III targets in different organisms reveal a wide range of putative targets. To our knowledge, no comparison of these datasets has been proposed yet and no clear conclusion can be proposed by comparing these studies except from the fact that RNase III has a large role in gene regulation in most organisms studied.

L165-173: Rather than enumerating the RNase III cleavages in different mRNAs, it would be more interesting to know (when possible) the consequences of these cleavages on the expression of the corresponding proteins. If I am correct, RNase III has been shown to control the expression of the ArfA protein providing an interesting mechanism to induce an alternative ribosome rescue pathway.

As recommended by the referee, section 2.2.3 was edited to highlight the role of Ec-RNase III in the positive regulation of ArfA abundance and in the repression of the accumulation of the osmoprotection associated factors proP, proU, betT and bdm.

L186-187: The authors state that RNase III is important for functions that are essential for virulence in different bacterial pathogens. They should replace “essential for virulence” by “involved in virulence”. In the cited references, there is no evidence for a role of RNase III in virulence.

Indeed, the role of RNase III is likely indirect, the sentence was corrected.

L188-194: The authors should clarify why “the second single-strand cleavage …cannot be observed by classic transcriptomic approaches”. In this study (ref 49), the authors attempted to identify trans sense-antisense duplexes, although their number was low.

The sentence was clarified as follows: “((i.e., the second single-strand cleavage is located in a second molecule thus a complex analysis is required to predict candidate dsRNA partners).”

L194-197: This sentence is not clear. What the authors mean by “explanation for the abundance of unsuspected RNase III cleavage sites”? Do the authors mean that transcriptomics data cannot detect all RNase III cleavage sites?

The sentence was corrected by the following clarification "(identified by transcriptomic approaches but not yet validated as direct targets)” as we were referring to the larger number of putative RNase III cleavage sites identified by transcriptomic approaches relative to validated cleavage sites.

L271: The authors could verify whether reference 84 is appropriate here.

The reference was moved as it may be confusing with the end of the sentence.

L289: Replace “DiGeroge” by “DiGeorge”.

The correction was introduced.

L325: This sentence is not correct. Indeed, RNase III binds double-stranded regions flanking the 16S (and not 16S/23S) and 23S (not 23S/5S) mature rRNAs.

The sentence was corrected.

L326-327: Precise that the additional 115 nt and 33 nt of the precursor are at the 5’ and 3’ ends respectively.

The sentence was corrected.

L331: Replace “decoupling” by “uncoupling”

The sentence was corrected.

L351: Reference 116 would be more appropriate than ref 122.

The reference was corrected.

L404: The “class I ArTL2” belongs to class II in Figure 4.

The sentence was corrected as AtRTL2 belongs to class II.

L450-452: This sentence is vague. What is the link between RNA interference and ribosome biogenesis?

The sentence was corrected as to not suggest a direct link between RNA interference and ribosome biogenesis.

L473-475: The authors might be more cautious with the interpretation of the data presented in reference 161. Indeed, this study has some shortcomings (the quality of the gels and Northern experiments is not satisfactory and the pre-rRNA have not been characterized) making it difficult to draw any conclusion.

The sentence was corrected to reflect a more hypothetical role of doxycycline as follows:

“Another report suggested that treatment of E. coli with doxycycline can lead to the accumulation of pre-rRNAs, hence the authors hypothesized that the binding of the drug could prevent the initial maturation of pre-rRNA by RNase III [165]"

L489-495: The authors indicate that RNase III produces a pre-23S rRNA four nucleotides shorter at the 5’ terminus than mature 23S rRNA suggesting that RNase III could be involved in alternative maturations pathways. However, this species has been found in both wild-type and RNase III deficient strain (ref 117) and the significance of this unusual 5’ terminus is not clear (see review of Srivastava and Schlessinger Annu. Rev. Microbiol. 1990).

The paragraph was edited to highlight the lack of information concerning this fragment.